# A Single Image Enhancement Technique Using Dark Channel Prior

**Cong Wang** [1,2], **Mingli Ding** [1,*], **Yongqiang Zhang** [1,*] and **Lina Wang** [1]

1. School of Instrumentation Science and Engineering, Harbin Institute of Technology, Harbin 150001, China; 19b901009@stu.hit.edu.cn (C.W.); 18b901007@stu.hit.edu.cn (L.W.)
2. Shanghai Institute of Satellite Engineering, Shanghai 200240, China
* Correspondence: dingml@hit.edu.cn (M.D.); zhangyongqiang@hit.edu.cn (Y.Z.)

**Abstract:** In this paper, we propose a novel single image enhancement technique for defogging by using dark channel prior. The traditional dark channel prior methods for defogging have problems of high time complexity, edge effect, and failure of dark channel prior. To overcome the problems of high time complexity and edge effect, firstly, a four-point weighting algorithm is proposed to estimate the atmospheric light value accurately, and the dark channel prior is used to estimate the rough transmittance. Then, the gray-scale image of the input image is used to refine the transmittance. After that, an atmospheric scattering model is designed to restore the fog-free image. To solve the problem that the dark channel prior can not process the high brightness area, a combination of edge detection and maximum inter-class variance is used to segment the sky area and non-sky area. Finally, the improved defogging method is used for processing the non-sky area, and the enhancement algorithm via sequential decomposition is used for handling the sky area. Extensive experiments show that the improved algorithm can not only reduce the time complexity, but also effectively improve the edge effect. At the same time, it can also solve the problem of failure of dark channel prior.

**Keywords:** dark channel prior; edge effect; four-point weighting; segmentation

## 1. Introduction

In the condition of fog and haze, the propagation of light will be affected by the scattering of suspended particles [1], which will attenuate features such as contrast and color of outdoor natural scenes captured by the image equipment. In the end, the image quality is severely degraded, and it will affect people's sight. Therefore, an image enhancement technique for processing of foggy images is a very practical requirement.

The techniques of removing fog from degraded images have a wide range of applications [2–4]. For this reason, many scholars have conducted long-term theoretical research and analysis on this direction, and many defogging algorithms have been proposed. From the perspective of image processing, existing mainstream defogging algorithms can be mainly divided into two categories: image enhancement technology-based methods and image restoration technology-based methods.

The image enhancement algorithms actually ignore the reasons for degradation of foggy images. They only consider the low brightness and low contrast characteristics of foggy images, and then directly enhance the information of interest in the fogging images. Common image enhancement algorithms include global or local histogram equalization algorithms [5] and defogging algorithms based on retinex theory [6]. The global histogram equalization algorithm has disadvantages of poor detail enhancement and color distortion. Although the local histogram equalization algorithm overcomes the shortcomings of the global histogram equalization algorithm, it has a large amount of calculations, so it is unsuitable for real-time processing systems. The defogging algorithm based on retinex theory works well for thin fog images, but when there is dense fog in the image, local

areas of fogging images are easily over-enhanced. In that case, the overall visual effect after defogging looks very unnatural.

Defogging methods based on image restoration can overcome the above disadvantages of enhancement technology-based methods. The authors of these methods conducted in-depth research on the specific foggy images, and proposed a series of image defogging algorithms based on atmospheric scattering models. According to the difference of additional information in the imaging scene, image restoration methods can be roughly divided into two types. In the first category, they first need to collect different weather images in the same scene, and then use these images as inputs [7] to implement their methods. The principle of these methods for image restoration is the differences between the input images. These methods have satisfactory defogging effect, but they need to acquire images under different weather, which results in poor real-time performance. In the second category, they take a fogging image as input, and prior knowledge or assumptions are used to perform image restoration. Literature [8] proposes an image defogging algorithm based on the prior condition, which is simple and fast, but local areas of restored images are too saturated. Literature [9] estimates the transmittance by assuming that the local transmittance is locally uncorrelated with the shading area in the scene. This algorithm has a better defogging effect on thin fog, but distortion will occur in dense fog areas. Tarel [10] assumes that the atmospheric dissipation function tends to the maximum value in a certain area, and then uses the median filtering to estimate its value. However, the median filter has a weak defogging ability in the areas of abrupt depth.

Aiming at overcoming the shortcomings of the above defogging algorithms, He [11] proposes a defogging algorithm based on the dark channel prior. This algorithm estimates the transmittance through the prior knowledge of dark channel, and uses soft matting technology to optimize the transmittance. It can achieve a good defogging effect for most natural images. However, the introduction of soft matting technology will increase the time complexity. At the same time, the algorithm is particularly prone to distortion when processing sky areas. Based on these shortcomings, a large number of researchers proposed a series of improved methods based on the dark channel prior. For example, in order to improve the speed of the defogging algorithm, a bilateral filtering instead of soft matting is used to refine the transmission. For defogging of the sky area, a tolerance mechanism [12] is proposed to correct the transmittance. Although these methods make up the shortcomings of He's algorithm to some extent, there is still room for improvement.

Specifically, traditional dark channel prior methods for defogging have problems of high time complexity, edge effect, and failure of dark channel prior. In this paper, we propose an improved image defogging method based on dark channel prior. For the first two disadvantages, we have the following improvements. First, the atmospheric light is accurately estimated by using a four-point weighting algorithm, then, guided filtering is used to repair the coarse transmittance obtained by using the dark channel prior. Finally, a fog-free image is restored through an atmospheric scattering model. Experimental results show that the improved algorithm can shorten the processing time and improve the edge effect. Furthermore, aiming at solving the problem of failure of dark channel prior in large areas with high brightness, we have the following improvements. First, the sky and non-sky areas are separated according to the method of edge detection. Then we use the enhancement algorithm via sequential decomposition to enhance the sky areas, and use the optimized dark channel prior method to handle the non-sky areas. Finally, we propose an improved image defogging method based on dark channel prior, which can effectively solve the problem of the failure of dark channel prior.

## 2. Dark Channel Prior Algorithm

### 2.1. Atmospheric Scattering Model

McCartney [13] simplified the atmospheric scattering model based on the Mie scattering theory, and then obtained the physical model of foggy images. The model consists of two parts, where the first part is called the incident light attenuation model, and the

second part is called the atmospheric light enhancement model. In the process of an object light reaching the observation device, the atmospheric light in other directions will enter the observation device, so that the atmospheric light obtained by the observation device is enhanced.

In the incident light attenuation model, the intensity of light reaching the observation device decreases exponentially as the depth of the scene increases, and the attenuation term of the incident light can be formulated as:

$$E_D(d, \lambda) = E_0(\lambda)e^{-\beta(\lambda)d} \tag{1}$$

where $E_D(d, \lambda)$ represents the light intensity of the object away from the target, $E_0(\lambda)$ denotes the intensity of the object light reflected by the target, $\beta(\lambda)$ represents the atmospheric scattering coefficient, and $d$ denotes the scene depth.

In the atmospheric light enhancement model, the intensity of light reaching the observation device increases exponentially as the depth of the scene increases, and the atmospheric light enhancement term can be formulated as:

$$E_A(d, \lambda) = E_\infty(\lambda)(1 - e^{-\beta(\lambda)d}) \tag{2}$$

where $E_A(d, \lambda)$ represents the intensity of atmospheric light away from the target, $E_\infty(\lambda)$ denotes the total intensity of unscattered atmospheric light, $\beta(\lambda)$ represents the atmospheric scattering coefficient and $d$ denotes the scene depth.

From the above analyses, the physical model of foggy images can be formulated as:

$$E(d, \lambda) = E_D(d, \lambda) + E_A(d, \lambda) \tag{3}$$

Substituting Equations (1) and (2) into Equation (3), we can get:

$$E(d, \lambda) = E_0(\lambda)e^{-\beta(\lambda)d} + E_\infty(\lambda)(1 - e^{-\beta(\lambda)d}) \tag{4}$$

If $I(x) = E(d, \lambda)$, $J(x) = E_0(\lambda)$, $t(x) = e^{-\beta(\lambda)d}$, and $A = E_\infty(\lambda)$, the model of foggy images can be simplified as:

$$I(x) = J(x)t(x) + A(1 - t(x)) \tag{5}$$

In the model, $x$ represents the two-dimensional coordinate value of the pixel, $I(x)$ denotes the intensity of the degraded image. $I(x)$ is a foggy image, and it also represents the input for image defogging algorithms. $J(x)$ means the intensity of the image before degradation, and it also represents the output of image defogging algorithms. $A$ represents the atmospheric light intensity, $t(x)$ denotes the transmittance and is used to describe the proportion of reflected light transmitted to the image acquisition device without being scattered on the surface of objects. Therefore, in order to obtain the defogged image $J(x)$, $t(x)$ and $A$ need to be estimated from $I(x)$.

### 2.2. Dark Channel Prior Theory

The phenomenon of fog usually appears in the scenes of outdoor and urban, so He [11] selected foggy outdoor scenes and urban scenes from the database. They randomly selected 5000 images and manually removed the sky area. In addition, it should be noted that He only paid attention to the fogging images in daytime scenes. These images were resized to $500 \times 500$ pixels, then they used a $15 \times 15$ window to calculate the dark channel values.

For the images excluding sky areas, the principle of dark channel prior is that at least one of the intensity values of the three channels of R, G, and B in the local area is very small, or even close to zero. For an input image $J$, the mathematical description of the dark channel prior $J^{dark}$ is as follows:

$$J^{dark}(x) = \min_{c \in \{r,g,b\}} \left( \min_{y \in \omega(x)} (J^C(y)) \right) \tag{6}$$

where $J^C$ represents the intensity values of the three color channels R, G, and B at location $y$, and $x$ is the center of the window $\omega(x)$.

In Equation (6), after two minimization operations, the dark channel value is obtained. The first step is to store the minimum value of the three color channels R, G, and B into a gray-scale image, which has the same size as the original image. In the second step, the window $\omega(x)$ is used to perform the minimum filtering operation on the gray-scale image obtained in the first step. The radius of the filter is determined by the window size.

There are three factors that result in a lower dark channel value: (1) The existence of shadows, such as the car shadows, buildings, the window of urban images, the shadow of large trees and leaves, and so on. (2) The color of objects, such as green grasses, forests, leaves, red flowers, blue water, etc. Due to the lack of other colors, i.e., the color is concentrated in a certain channel, which causes the values of other channels to be relatively low, a dark channel is generated. (3) Black objects, such as black stones, tree trunks, etc. In the real world, most of the objects have color information or have shadow areas, thereby ensuring the universality of the dark channel prior.

### 2.3. Disadvantages of Defogging Using Dark Channel Prior

In the soft matting algorithm, the matting model can be formulated as:

$$I = Fa + B(1 - a) \tag{7}$$

where $F$ and $B$ represent foreground and background areas, respectively, $a$ is the transparency of the foreground.

Assuming that the refined transmittance is $t(x)$, and $t(x)$ and $t'(x)$ are re-written to $t$ and $t'$, the objective function of soft matting can be formulated as:

$$E(t) = t^T Lt + \lambda(t - t')^T(t - t') \tag{8}$$

where $t^T Lt$ and $\lambda(t - t')^T(t - t')$ are the smoothing term and data term, respectively, $\lambda$ is a weighting factor, $L$ is a matte Laplacian matrix, and the elements $(i, j)$ in $L$ are defined as:

$$\sum_{k(i,j)\in\omega_k} (\theta_{ij} - \frac{1}{|\omega_k|}(1 + (I_i - u_k)^T(\sum_k + \frac{\gamma}{|\omega_k|}U_3)^{-1}(I_k - u_k))) \tag{9}$$

where $u_k$ and $\sum_k$ are the mean and variance of the matrix in the window $\omega_k$. $I_i$ and $I_j$ denote the color of the input image $I$ at pixels $i$ and $j$, respectively. $\theta_{ij}$ represents the Kronecker function, $U_3$ represents a $3 \times 3$ identity matrix, $\gamma$ represents a normalization parameter and $|\omega_k|$ denotes the number of pixels in the window $\omega_k$.

We can get the optimal $t$ by solving the sparse linear system of Equation (10):

$$(L + \lambda U)t = \lambda t' \tag{10}$$

Using this matting algorithm to repair the acquired transmission can achieve satisfactory results, but the algorithm involves the reversible operation of a large matrix, which takes up more than 95% time of the defogging algorithm. Aiming at using the soft matting technology to refine the transmittance results in a high time complexity, we use guided filtering instead of soft matting to refine the transmittance, which can not only maintain the edge information, but also shorten the defogging time.

Dark channel prior defogging algorithms use minimum filtering strategy in the process of obtaining transmittance. Then, using local block area $\omega(x)$ as a template to perform point-by-point calculations, and take the minimum value of the R, G, and B channels in the block area as the dark channel value of the center pixel. Finally, the transmittance is estimated by the dark channel prior theory. The local filtering process shows that the transmittance of each pixel in the image is obtained by estimating the transmittance of the pixels in its adjacent area, but it is not the true transmittance value. Figure 1 shows local filtering for smooth depth of field and depth-changing edges. Among them, $3 \times 3$ local block is the filter window $\omega(x)$, the dark box area represents its central pixel, the A-side area is a distant area with low pixel intensity, and the B-side area is a near-field area with high pixel intensity.

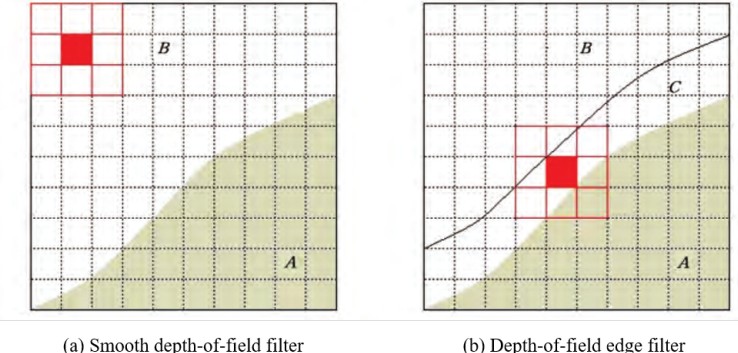

(a) Smooth depth-of-field filter       (b) Depth-of-field edge filter

**Figure 1.** The figure of local filtering for flat depth-of-field areas and depth-of-field changing edges.

As shown in Figure 1a (left side), when the filtering window $\omega(x)$ is a scene with a smooth depth of field (the filtering window is the highlighted area of close scene B in the figure), the dark channel value of the central pixel can be accurately obtained through the minimum filtering. The transmittance of each pixel in the window can take approximately the same value, and it can achieve the desired defogging effect. As shown in Figure 1b (right side), when the depth of filter window $\omega(x)$ changes (the boundaries of A and B regions in the filter window), the dark channel value of the central pixel obtained by the minimum filter will become small, and the transmittance value of the pixels in the window are not similar. In this way, a dark transition zone (C region as shown in Figure 1b) will appear between A and B regions after filtering, resulting in the blur of depth-changing edges in the defogged images, thereby causing obvious edge effects. Aiming at solving the above-mentioned problems, we first use a four-point weighting algorithm to accurately estimate the atmospheric light value. Then, under the premise that the atmospheric light value is accurately estimated, the dark channel prior is used to obtain an accurate transmittance value. At the same time, we use the gray-scale image of the input image as the guide image, which can better maintain the edge of the image. Thus, the edge effect of defogged images can be improved.

Furthermore, the failure of the dark channel prior theory generally occurs in the areas where the intensity of the scene is close to the intensity of the atmospheric light, such as gray and white scenes, sky, reflective water surface, and other large areas with high brightness. This is because the true transmittance value of such areas is relatively high, and the visual effect is basically the same under fog-free and foggy weather. However, in the dark channel map, the gray value of the dark channel map is very high because there are no low pixel values in such areas. According to the transmittance formula, the estimated transmittance of this type of area is very small, which is different from the true transmittance of this area. As shown in Figure 2a (left side), the color of foggy images in large areas (e.g., the sky area) is similar to the visual effect of the atmospheric light intensity. As shown in Figure 2b (right side), after defogging based on the dark primary color principle, there is a phenomenon that the defogging is not obvious and the visual effect is poor in the sky area. In response to the above-mentioned problem that knowledge of dark channel prior for large areas of high brightness will fail, we first use a combination of edge detection and maximum inter-class variance to separate the sky and non-sky regions. Then, an enhancement algorithm based on sequential decomposition is used on the sky area; thus, the fog in large areas with high brightness can be removed thoroughly.

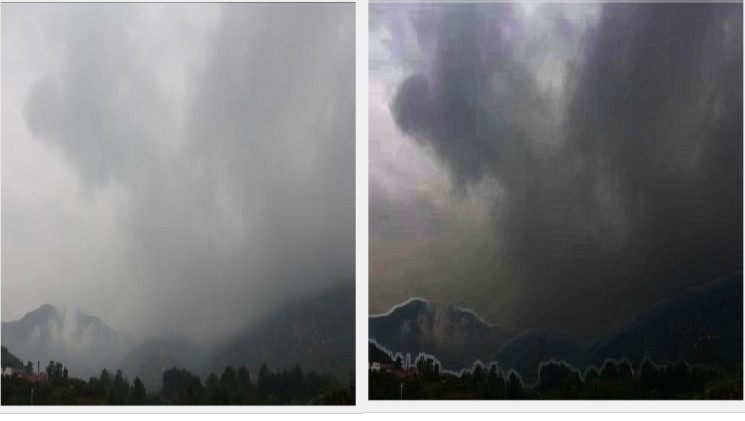

(a) Mostly foggy sky            (b) Sky without fog illustration

**Figure 2.** Comparisons of dark channel failure examples.

## 3. Improved Image Defogging Method Based on Dark Channel Prior

*3.1. Improvement of High Time Complexity and Edge Effects of Dark Channel Prior Algorithm*

Aiming at solving the problems of high time complexity and edge effect of traditional dark channel prior algorithms, in this paper, we propose an improved method. First, a four-point weighting algorithm is used to estimate the atmospheric light value accurately. Then, the rough transmittance is estimated by using the dark channel prior principle. In order to avoid edge effect and maintain the defogging edge, a guided filter is introduced to refine the coarse transmittance map. Finally, the fog-free image is restored by the atmospheric scattering model. The detailed improvements are described as below.

### 3.1.1. Atmospheric Light Value Estimation Based on Four-Point Weighting

The estimation of the atmospheric light value in He's algorithm [11] is equal to 0.1% of the gray value in the global dark channel image, which will get poor results when the image contains a large bright area. In order to accurately estimate the atmospheric light value $A$, this paper proposes a four-point weighting algorithm to find the optimal atmospheric light value.

The atmospheric light value is obtained in the area with the highest fog concentration in the foggy image, where the area is generally defined as a rectangular area. For an image, when the fog concentration in the area is higher, the pixel value becomes higher, but the difference between the pixels becomes smaller, so the difference between the mean value and the standard deviation of the pixel becomes larger.

To estimate the atmospheric light value $A$ of an inputted image, as shown in Algorithm 1, we divide the input image $I$ into four regions (i.e., $i_n$, $n = 1, 2, 3, 4$) with the same size, and the difference of each region is calculated by Equation (11):

$$S(n) = M(n) - D(n) \tag{11}$$

where $M(n)$ and $D(n)$ represent the mean and variance of a region, respectively, $n = 1, 2, 3, 4$ represents one of the four regions.

We select the region with the largest difference in four regions. Then, the selected region is used as the new input image to repeat the above processing until the preset requirements $Pr$ are met, and the final selected region is defined as $Y(x)$. In this paper, the preset requirements $Pr$ are defined as: (1) the size of the segmented area is less than 1/16 of the original input image; (2) there are two regions with the largest difference and the average of two regions is the same; (3) there are three regions with the maximum difference in four regions.

In order to obtain the atmospheric light value $A$ in the region $Y(x)$, we first calculate the average pixel value $M(n)$ of $Y(x)$. Then, the pixels in $Y(x)$ are divided into two parts according to the pixel value. All pixels larger than $M(n)$ are called bright pixels, and all

pixels smaller than $M(n)$ are called dark pixels. The number of bright and dark pixels are denoted as $N_b$ and $N_d$, respectively. Finally, we find the maximum values $A_b$ and $A_d$ of dark channel in the light and dark areas, respectively, and it is assumed that $A_b$ and $A_d$ are obtained at $Y(n_1)$ and $Y(n_2)$. The atmospheric value $A$ is calculated as follows:

$$
\begin{cases}
A = W_b * Y(n_1) + W_d * Y(n_2) \\
W_b = \dfrac{N_b}{L * W} \\
W_d = \dfrac{N_d}{L * W} \\
W_b + W_d = 1
\end{cases}
\tag{12}
$$

where $L * W$ denotes the size of the region, and the value of atmospheric light $A$ is estimated by weighting $Y(x)$. When $N_b > N_d$, $Y(n_1)$ dominates the atmospheric light value A. When $N_b < N_d$, $Y(n_2)$ dominates the atmospheric light value A. $Y(n_1)$ and $Y(n_2)$ influencing $A$ together makes the obtained atmospheric light value $A$ reasonable. Thus, using the atmospheric light value $A$ to remove fog from an image makes the defogged image consistent with the human visual system.

---

**Algorithm 1** Four-point Weighting Algorithm

---

**Require:** Input image $I$, Preset requirements $Pr$
    **while** !Pr **do**
        $i_n = divide(I), n = 1, 2, 3, 4$
        **for** $m < n$ **do**
            $S(n) = M(n) - D(n)$
        **end for**
        $Sel = max(S(n))$
        $Sel\_index = index(max(S(n)))$
        $I = i(Sel\_index)$
    **end while**
    $Y(x) = Sel$
    $M(n) = mean(Y(x))$
    $N_b = num(pixel > M(n))$
    $N_d = num(pixel < M(n))$
    $W_b = \frac{N_b}{L*W}$
    $W_d = \frac{N_d}{L*W}$
    $A = W_b * Y(n_1) + W_d * Y(n_2)$
**Output:** Atmospheric Light $A$

---

### 3.1.2. Refinement of Coarse Transmittance Based on Guided Filtering

After obtaining the rough transmittance by using the dark channel prior, we use guided filtering instead of soft matting to refine the transmittance. Guided filtering has a filtering feature of locally linear smooth-preserving edges, which can be used to repair the rough transmittance obtained by the dark channel prior. The principle is to filter the input image through a guide image, and the output image can fully obtain the detailed changes of the guide image while retaining the overall characteristics of the input image. Compared with soft matting, guided filtering greatly improves the efficiency of the algorithm without affecting the visual effect.

Guided filtering assumes that there is a local linear relationship between the guided image $I$ and the output image, that is:

$$
q_i = a_k I_i + b_k (i \in w_k) \tag{13}
$$

where $w_k$ is a square window, $k$ is the center pixel, $a_k$ and $b_k$ are linear factors in the window and constant in the window, respectively. Equation (13) represents a linear transformation centered on $k$ in the image $I$. This linear transformation ensures that when the guiding image $I$ in the window has edges, the output image will have corresponding edges. The linear

regression method is used to search the window coefficients $a_k$ and $b_k$ with the smallest window cost, and then the local linear filtering with the smallest difference between the input image and the output image is used by the guide image:

$$E(a_k, b_k) = \sum_{i \in w_k} \left( (a_k I_i + b_k - p_i)^2 + \theta a_k^2 \right) \tag{14}$$

where $\theta$ is an hyper-parameter, and its purpose is to prevent the value of $a_k$ from being too large. The window coefficient can be obtained by linear regression analysis in the literature [14]:

$$a_k = \frac{\frac{1}{|w|} \sum_{i \in w_k} (p_i I_i - u_k p_k')}{\omega_k^2 + \theta} \tag{15}$$

$$b_k = p_k' - a_k u_k \tag{16}$$

In Equations (15) and (16), $|w|$ is the number of pixels in the window $w_k$. $u_k$ and $\omega_k$ are the mean and variance of the guide graph in the window $w_k$, and $p_k'$ is the mean of the input image in the window $w_k$. As a result that point $i$ may be contained in different windows $w_k$ and the values of $a_k$ and $b_k$ are different, it is necessary to average the values of $a_k$ and $b_k$ involving all points $i$. Therefore, the final expression of the guided filtering is obtained:

$$q_i = \frac{1}{|w|} \sum_{k \in w_k} (a_k I_i + b_k) = a_i' I_i + b_i' \tag{17}$$

Through the above analysis, the specific steps of using the guided filtering to optimize the transmittance are as follows: First, we use the rough transmittance map estimated by the dark channel prior algorithm as the input image $p$. Then, because the guide image $I$ needs the edge information to reflect the depth information, we use the gray-scale image of the original image as the guide image $I$.

After obtaining the improved atmospheric light value and transmittance, the atmospheric scattering model is used as a physical model for foggy image degradation, and then this model is used to obtain a defogged image. A comparative analysis is performed by using an unimproved dark channel prior algorithm and the improved method proposed in this paper, as shown in Figure 3.

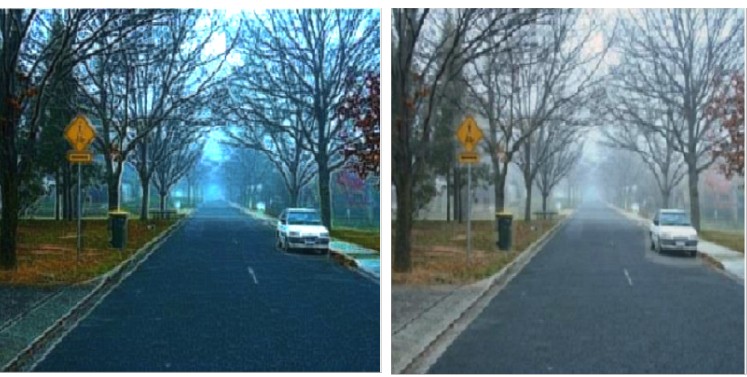

(a) Dark channel dehazing    (b) Improved method of this article

**Figure 3.** Comparison of the defogging effect of two methods.

As we can see from Figure 3a, after using the dark channel prior to removing the fog, there are obvious edge effects on the tree periphery, the car outline, the signboard periphery, and the road side. At the same time, there is also obvious color cast. However, after using the improved method of this paper to remove the fog, it can be seen from Figure 3b, the edge effects of trees, cars, signs, and roadsides have been greatly improved, and the color of the image is natural.

### 3.2. Treatment of Dark Channel Prior Failure Areas

The sky region does not conform to the principle of dark channel prior, so the transmission of the sky region is mis-estimated, causing halo distortion in sky regions. As a result, various defogging algorithms based on sky region segmentation are emerging. For example, Wang Guangyi [15] uses gradient threshold and region growth algorithm to obtain the unicom region, and then identifies the sky region based on the pixel brightness threshold of the unicom region. However, the phenomenon of missed detection in the sky region and the time-consuming calculation of the unicom region occurred. Aiming at solving these problems, in this paper, we propose an improved method as follows. First, an algorithm based on a combination of edge detection and maximum inter-class variance is used to separate the sky and non-sky areas. Then, the enhancement algorithm via sequential decomposition is used for the sky area, and the optimization method of Section 3.1 is used for the non-sky area. Finally, the enhanced sky and non-sky areas after defogging are fused to obtain a clear defogged image.

#### 3.2.1. Sky Region Segmentation Based on Edge Detection and Maximum Inter-Class Variance

In this section, we propose a method based on edge detection and maximum inter-class variance to optimize the sky region segmentation algorithm. The specific steps of sky segmentation in this paper are as follows: (1) convert RGB images into gray-scale images with the goal of retaining more edge information; (2) use Sobel to calculate the gradient information of the gray-scale image; (3) the gradient information is distinguished according to the gradient threshold and brightness threshold. The gradient threshold in this paper is set to 0.83. The brightness threshold is obtained by maximum inter-class variance. The segmentation image is shown in Figure 4.

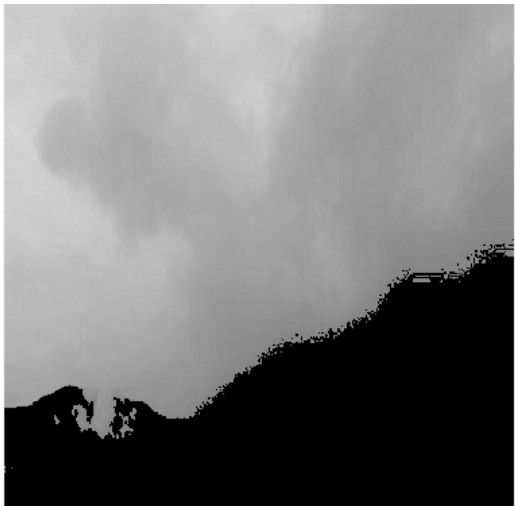

**Figure 4.** The sky segmentation example.

#### 3.2.2. Sky Enhancement Algorithm Based on Sequence Decomposition

In this section, we propose a sky enhancement algorithm based on sequential decomposition for the sky area. The enhancement algorithm based on sequence decomposition performs decomposition based on the Retinex model in continuous sequences, and continuously estimates a smooth illumination and a reflectance. After getting the illumination and reflectance, the light layer is used to produce an enhanced effect, and the specific steps are as follows:

Step 1: Estimate the initial illumination $L'$, and convert the RGB image to a YUV image. In the YUV space, set the $Y$ channel of the input image to the initial illumination $L'$.

Step 2: Use Equation (18) to estimate the optimal illumination $L$.

$$argmin||L - L'||_F^2 + a||\Theta L||_1 \tag{18}$$

where $||L - L'||_F^2$ represents the fidelity of the initial illumination $L'$ and the optimal illumination $L$, and $\Theta$ represents the first order differential operator.

Step 3: Calculate the correlation weight matrix $W$ of the input image and the adjustment gradient $G$ of the input image.

$$W = \frac{1}{|\Theta s| + \omega} \tag{19}$$

where $\omega$ is a threshold value for eliminating small gradient.

Step 4: Use Equation (20) to estimate the optimal reflectance $R$.

$$argmin_R||R - S/L||_F^2 + \beta||W * \Theta R||_F^2 + w||\Theta R - G||_F^2 \tag{20}$$

where $||R - S/L||$ represents the fidelity of $R$ and $S/L$, and $||W * \Theta R||_F^2$ represents the spatial smoothness on the enhanced reflectance $R$. The purpose of $||\Theta R - G||_F^2$ is to reduce the gradient of the reflectance $R$ and the observed image $S$.

Step 5: Use Equation (21) to enhance the image.

$$S' = R * L'^{\frac{1}{r}} \tag{21}$$

The improved algorithm is compared with the sky defogging algorithm based on cost function [16], sky defogging algorithm based on tolerance mechanism [12] and defogging algorithm [15] based on sky segmentation, and Figure 5 illustrates the comparison figures.

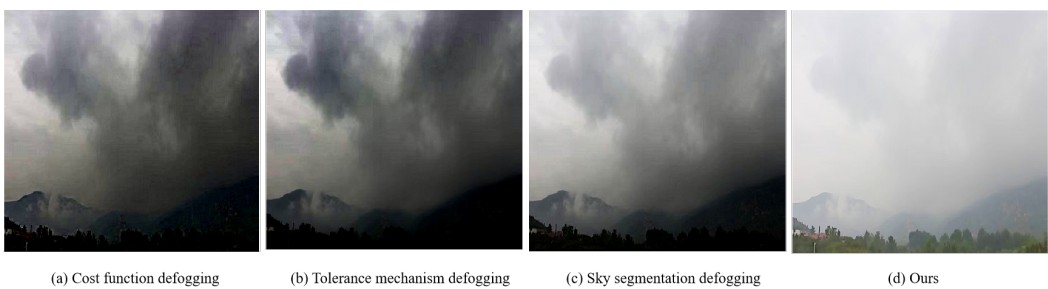

　(a) Cost function defogging　　　(b) Tolerance mechanism defogging　　　(c) Sky segmentation defogging　　　(d) Ours

**Figure 5.** Comparison results of different defogging algorithms, including cost function defogging algorithm [16], tolerance mechanism defogging algorithm [12], sky segmentation algorithm [15], and our proposed method.

## 4. Experimental Results And Analysis

We verify the effectiveness and performance of the improved algorithm through experiments, simulating and analyzing the subjective visual effect and objective quantitative results of the defogged image. Following [11], 5000 images from the dataset are used in our experiments, which are captured from foggy outdoor and urban scenes. All our experiments are simulated in Matlab (2017a), and the operating environment is 3.20 GHz Intel Core (i5-4460) CPU, 4G memory, 64-bit win10 system. In Equation (18), we set $a = 0.001$. In Equation (20), we set $\beta = 0.007$, $\omega = 0.016$. Under normal circumstance, such parameter settings can obtain satisfactory defogged results.

### 4.1. Subjective Visual Evaluation

In order to verify that the improved algorithm proposed in this paper has greatly improved the edge effect, an experimental simulation is performed. The obtained defogged restoration results are compared with defogged results of He's algorithm. Figure 6 illustrates the defogged images obtained by two defogging algorithms in different foggy scenes.

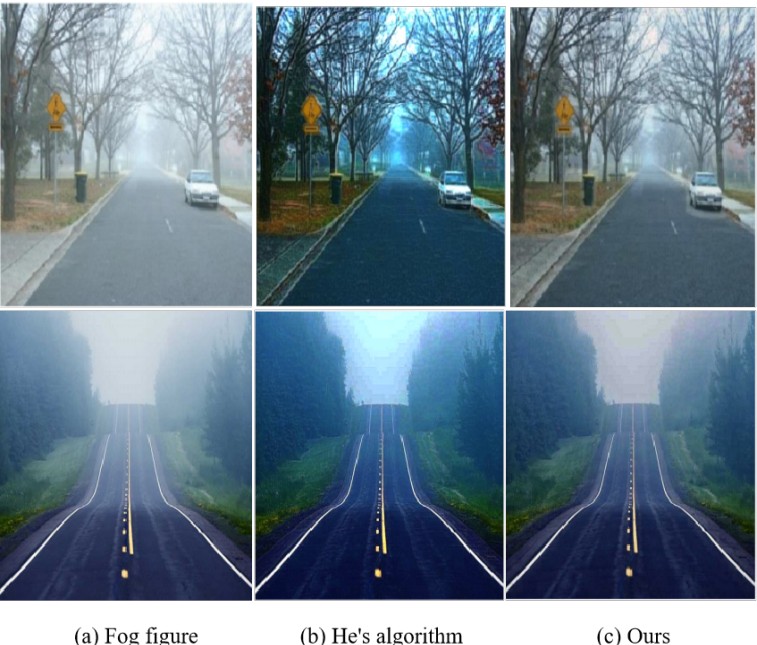

　　　(a) Fog figure　　　　　(b) He's algorithm　　　　　(c) Ours

**Figure 6.** Comparison with He's algorithms [11] in different scenarios.

　　　The first image and the second image in Figure 6b are defogged images obtained by using He's algorithm [11]. The overall color of images has a large chromatic aberration phenomenon. At the same time, an obvious edge effect appears in the tree outlines, the car peripheries, and on the signboards and the roads. This is mainly because the atmospheric light value in He's defogging algorithm is not appropriate. When there is a large white object in the image, the method incorrectly estimates the atmospheric light value. This will cause an inaccurate transmittance estimation and eventually cause an edge effect.

　　　The first image and the second image in Figure 6c are defogged images obtained by using our improved method. It can be seen that the defogged images we obtained not only make the color of the overall image very real, but also greatly improve the edge effects on the sides of trees, cars, signs, and roads. This is mainly because our improved method estimates the atmospheric light value by using a four-point weighting algorithm. This algorithm can deal with the situation that the estimation of the atmospheric light value is not accurate.

　　　In summary, the improved algorithm we proposed can subjectively improve the edge effect. Through the experimental simulation, it is verified that the improved method can achieve better defogged results on large areas with high brightness where the dark channel prior fails than He's method. Comparing our defogged images with current algorithms demonstrates that our method performs better on large areas with high brightness than other methods, such as defogging method based on cost function [16], defogging method based on tolerance mechanism [12], and defogging method based on sky segmentation [15]. Figure 7 illustrates the defogged images from four defogging algorithms.

　　　Furthermore, the two scenes in Figure 7 are a large sky area and a large reflective water surface. The first and second images in Figure 7b are defogged images obtained by using defogging algorithm based on the cost function [16]. It can be seen from the two images that the fog becomes darker, resulting in less detailed information in the image after defogging. This is mainly because the defogging method based on the cost function does not emphasize the refinement of transmittance.

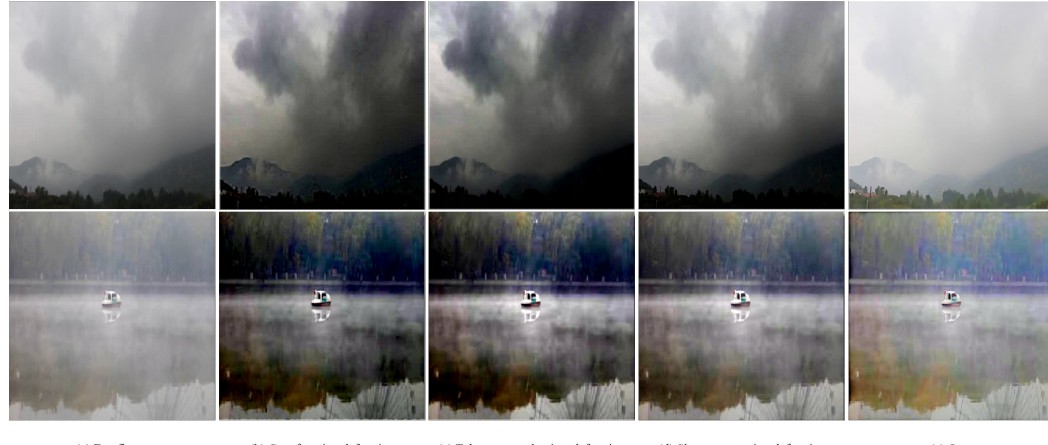

<div align="center">(a) Fog figure      (b) Cost function defogging      (c) Tolerance mechanism defogging      (d) Sky segmentation defogging      (e) Ours</div>

**Figure 7.** Comparison of the effect of different defogging algorithms, including cost function defogging algorithm [16], tolerance mechanism defogging algorithm [12], and sky segmentation algorithm [15], on defogging in bright areas.

The first picture and the second picture of Figure 7c are defogged images obtained by using defogging algorithm [12] based on the tolerance mechanism. It can be seen from the two pictures that the fog in the sky area of the first picture is still not removed. It shows large distortion on the water surface and the lake in the second image. This is mainly because the essence of the defogging algorithm [12] based on the tolerance mechanism uses a controllable parameter to distinguish the sky area from the non-sky area.

The first image and the second image in Figure 7d are defogged images obtained by using segmentation algorithm [15]. It can be seen that most of the fog has been removed, but the thick fog has not been removed. This is mainly because the idea of this algorithm is to segment the sky and non-sky, and then perform different processing on the transmittance of the two parts. Among them, the transmittance of the sky region is set to a constant. This approach also makes the transmittance estimation of the sky area inaccurate, which will cause undesired defogging in the sky area.

The first and second images in Figure 7e are defogged images obtained by using our improved method. It can be seen from the first image that not only the dense fog in the original image is removed, but also the small villages next to the mountains can be seen. From the second picture, we can see that the fog on the reflective water surface has been removed. This is mainly because the defogging method we used for large areas of high lightness is not dark channel prior, but a new method. First, the sky segmentation algorithm is used to separate the sky region from the non-sky region. Then, an optimized dark channel prior algorithm is used for the non-sky region. Finally, an enhancement algorithm based on sequential decomposition is used for the sky region. Our method does not perform noise cancellation and sky enhancement separately, but processes them at the same time. In this sequence of suppressing noise, we perform spatial smoothing for each component, and use the weighting matrix to suppress noise.

Based on the above described subjective analysis, the sky enhancement algorithm based on sequential decomposition can effectively overcome the problem of failure of defogging in large areas with high lightness. Our method can not only remove the fog subjectively, but also enrich the details of the fogged image.

### 4.2. Quantitative Comparisons

In order to fully evaluate the defogging effect, PSNR and SSIM are used to quantify the defogged results of different algorithms. Taking Figures 6 and 7 as examples, the evaluation results are shown in Tables 1 and 2, respectively. Moreover, we also report the evaluation results on the test set in Table 3.

Peak Signal-to-Noise Ratio (PSNR) measures the image quality by calculating the pixel error between the image to be evaluated and the reference image. The larger the

PSNR value, the smaller the distortion between the image to be evaluated and the reference image, and the better the image quality. The definition of PSNR can be formulated as:

$$PSNR = 10 * log_{10} \frac{MAX_I^2}{MSE},$$ (22)

where $MAX$ denotes the maximum pixel value of the image $I$, $MSE$ represents the mean square error between two images.

Structural SIMilarity (SSIM) is an objective criterion for image quality that meets the characteristics of the human visual system. The larger the value of SSIM, the better it can reflect people's subjective feelings. The formulation of SSIM can be defined as:

$$SSIM(x,y) = \frac{(2u_x u_y + c_1)(2\delta_{xy} + c_2)}{(u_x^2 + u_y^2 + c_1)(\delta_x^2 + \delta_y^2 + c_2)},$$ (23)

where (x,y) denotes two samples, $u_x$ denotes the mean value of $x$, $u_y$ denotes the mean value of $y$, $\delta_x$ denotes the variance of $x$, $\delta_y$ denotes the variance of $y$, $\delta_{xy}$ denotes the co-variance of $x$ and $y$, and $c_1$ and $c_2$ are constants.

**Table 1.** Quantitative results of He's algorithm [11] and our proposed methods in the images of Figure 6.

| Methods | Scenario 1 | | | Scenario 2 | | |
|---------|------|------|----------|------|------|----------|
| | PSNR | SSIM | Run Time | PSNR | SSIM | Run Time |
| He's [11] | 10.65 | 0.44 | 1.8 | 16.5 | 0.79 | 1.7 |
| Ours | 15.15 | 0.63 | 1.5 | 17.19 | 0.81 | 1.4 |

As can be seen from Table 1, in scene 1, the SSIM value obtained by using the algorithm of this paper is 0.19 larger than that obtained by He's defogging algorithm, which means that the defogged image obtained by using the algorithm of this paper can better reflect people's subjective feelings. The PSNR value obtained by using the algorithm of this paper is 4.50 larger than that obtained by He's defogging algorithm. That is, the defogging effect obtained by using the improved algorithm of this paper has smaller pixel error and better image quality than the first image of Figure 6a. The time obtained by the algorithm of this paper is 0.3s less than the time obtained by He's defogging algorithm. In He's defogging algorithm, he used the soft matting to refine the transmittance. Although this matting algorithm can be used to repair the obtained transmittance and achieve satisfactory results, the algorithm involves the reversible operation of large matrices, resulting in high time complexity. In this paper, guided filtering instead of soft matting can not only refine the transmittance, but also shorten the image processing time. A similar improvement can be found in scene 2.

In summary, no matter from the SSIM, PSNR, or time, we can objectively verify that the proposed method can improve the edge effect and shorten the image processing time.

As can be seen from Table 2, the algorithm in this paper is quantitatively compared with three classic sky defogging algorithms. For scene 1, the improved algorithm we proposed is 5.45 larger in PSNR than the defogging algorithm [16] based on cost function, 5.72 larger than the PSNR of the defogging algorithm based on tolerance [12], and 3.58 larger than the PSNR of the defogging algorithm [15] based on the sky segmentation. Therefore, whether it is from the previous subjective analysis or objective index analysis, it can explain that the defogged images obtained by our improved method have less distortion and better image quality. For scene 2, the improved algorithm we proposed is 0.28 larger in SSIM than the defogging algorithm [16] based on cost function, 0.28 larger than the SSIM of defogging algorithm [12] based on the tolerance mechanism, and 0.11 larger than the SSIM of defogging algorithm [15] based on the sky segmentation. Therefore, it is verified again from objective indicators that the defogged image obtained by our improved algorithm can better reflect human subjective feelings.

**Table 2.** Quantitative results of other defogging methods and our proposed method in the images of Figure 7.

| Methods | Scenario 1 | | Scenario 2 | |
|---|---|---|---|---|
| | PSNR | SSIM | PSNR | SSIM |
| Cost function defogging method [16] | 13.25 | 0.75 | 8.79 | 0.46 |
| Tolerance mechanism defogging method [12] | 12.98 | 0.77 | 9.15 | 0.46 |
| Sky segmentation defogging method [15] | 15.12 | 0.90 | 11.37 | 0.63 |
| Ours | 18.70 | 0.92 | 13.32 | 0.74 |

**Table 3.** Quantitative results of other defogging methods and our proposed method in the test set.

| Method | PSNR | SSIM |
|---|---|---|
| Cost function defogging method [16] | 11.58 | 0.66 |
| Tolerance mechanism defogging method [12] | 13.16 | 0.72 |
| Sky segmentation defogging method [15] | 15.06 | 0.88 |
| Ours | 19.20 | 0.91 |

Moreover, following [11], 5000 images from the dataset are used in our experiments. Here, we use 4000 images to select the hyper-parameters in our proposed method, and use the remaining 1000 images to verify the effectiveness of our method. The evaluation results are shown in Table 3. From Table 3, we can see that our method surpasses other methods by a large margin in PSNR and SSIM respectively, which further demonstrates the effectiveness of our proposed methods.

## 5. Conclusions

Based on the traditional dark channel prior model, this paper proposes several improvements for three shortcomings (i.e., high time complexity, edge effect, and failure of dark channel prior) of He's algorithm. First, a four-point weighting algorithm is used to accurately estimate the atmospheric light value, and guided filtering is introduced instead of the more complex soft matting algorithm to refine the coarse transmission map. In this case, not only the processing speed of the algorithm is improved, but also the edge effect is effectively improved. Then, the sky area does not conform to the dark channel prior principle, which can cause halo distortion in sky areas. An algorithm combining edge detection and maximum inter-class variance is proposed to separate the sky area from the non-sky area, and then an enhanced algorithm via sequential decomposition is applied to the sky area. Experimental results show that the improved methods not only improve the visual effect of the defogged image subjectively, but also objectively evaluate that the defogging effect of our proposed method is more thorough and the recovered details are more abundant than other methods.

**Author Contributions:** Methodology, C.W., Y.Z. and L.W.; Supervision, M.D. All authors have read and agreed to the published version of the manuscript.

**Funding:** This work is supported by the China Postdoctoral Science Foundation (Grant No. 259822), the National Postdoctoral program for Innovative Talents (Grant No. BX20200108), the National Science Foundation of China (Grant No. 61976070), and the Science Foundation of Heilongjiang Province (YQ2020F005).

**Institutional Review Board Statement:** Not applicable.

**Informed Consent Statement:** Not applicable.

**Data Availability Statement:** Data is contained within the article.

**Conflicts of Interest:** We would like to note that in the manuscript entitled "A Single Image Enhancement technique using Dark Channel Prior", no conflict of interest exits in the submission of this manuscript, and manuscript is approved by all authors for publication.

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
