# Peer review of "A Single Image Enhancement Technique Using Dark Channel Prior"

_applsci, doi:10.3390/app11062712_

Round 1

Reviewer 1 Report

The paper proposed enhancement method for defogging in comparison with He's algorithm. The 4-point weighting algorithm was presented clearly (from line 220). It should be written in mathematical or algorithm form in stead of using "steps". The sky region segmentation was presented and I doubt the effectiveness of the algorithm with different input images. The gradient threshold 0.83 may be used for specific inputs. The experiments of the proposed method should be extensively carried out. How many images were used in the experiments? Tables 1 and 2 showed the comparisons of 4 specific examples. The paper contains some typos and should be revised carefully. For example: scenarios 1, 2 in Table 1, 2; ?=0.65 (line 300) Cite the methods in Fig. 7 and Table 2.

Author Response

Please see the attachment, thanks.

Reviewer 2 Report

The paper proposes various improvements on previous algorithms intended to defog hazy images, an area with a lot of activity. Basically, they focus on the upgrade of the algorithm proposed by Kaiming He, analyzing specific circumstances where it might fail.

I think that the paper is properly presented, with enough original results, even though the improvements are not that large. However, the analysis and proposals are interesting for the community working in this field and will help with the advancement for better techniques.

Then, my decision is that the paper can be accepted with minor corrections, as follows:

Line 32: typo in the reference.

Line 59: “of the scene in the scene” repeated

Section 4.1.1.: definitions for PSNR and SSIM

Table 1 and Table 2: abuse of decimals since probably significant is only the first one

Conflict of interest Section: to be completed

Line 507: Ref. [25] not properly cited. This is important since it is cited many times in the text.

Author Response

Please see the attachment, thanks.

Reviewer 3 Report

Reasonably well written, few language edits required...

Reviewer 4 Report

The authors propose an image defogging algorithm based on dark channel prior. They build on He’s haze removal method, but address several limitations, such as time complexity, edge effects and large bright areas. The paper explains the proposed method quite well (the steps of the method are detailed, and the authors present the reasons for introducing each new processing). I believe that the details are sufficient to reproduce the method. The authors present both a qualitative and a quantitative evaluation. However, there are several concerns regarding the paper:

  • The paper is full of English mistakes and strange phrases, making it difficult to follow. In my opinion, the paper should be proofread by a professional editing company or by a Native English Speaker. I give only a few examples, but I must stress out that there are many other mistakes:
    • “Image enhancement technology has been widely studied in recent decades, in this paper, we propose …” - there should be a period after “decades”, and a new phrase should begin
    • “To solve the problem of dark channel prior is not applicable to process large areas with high brightness, …” - sentence makes no sense
    • “a wide range of application” - applications
    • “These images were redefined as 500*500 pixels” – resized
    • “Local filtering of smooth depth for field and depth-changing edges.” – depth of field
    • “For example, gray and white scenes, sky, reflective water surface and other large areas with high brightness.” - this sentence has no verb
    • and so on…
  • In my opinion, a quantitative evaluation on only 4 images is not relevant. I believe that at least 50-100 images should be processed with several defogging methods, computing the proposed metrics (PSNR and SSIM)
  • I believe that a discussion regarding the possibility of accelerating the method (for example, using GPGPU) would be welcome, since a performance of approx. 1.5 seconds seems quite low

Particular comments:

  • In Algorithm 1 and its description, I do not understand how Y(m) and Y(n) are determined. I also believe that using M and M(n) for representing different things might be confusing. I also do not understand how the preset requirements (1),(2) and (3) were chosen.
  • Every acronym should be explained (e.g. PSNR, SSIM)

Round 2

Reviewer 1 Report

It was stated that the authors carried out the experiment with 5000 images. However, only 4 example results were mentioned in Figures 6, 7 and Tables 1, 2.
How can the authors claim that their method is better than others without showing the results of all the tests? 

Reviewer 4 Report

The authors have addressed my comments. However, in the received manuscript, all the references to the bibliography and to the figures are missing (probably the pdf was not properly built from the latex file). Moreover, there are still a lot of English mistakes and phrases that are difficult to follow. I am not a native English speaker, but most of these mistakes are obvious. I do not understand how a native speaker would have missed them. Here are some examples:

  • “To solve the problem that the dark channel prior can not to process the high brightness area” – CANNOT PROCESS.
  • “The image enhancement algorithms actually ignore the reasons for degradation of foggy images. It only considers the low brightness and ….” – THEY only CONSIDER (because the phrase talks about algorithms, which are in plural form)
  • “Due to the lack of other colors, i.e. the color is concentrated in a certain channel, which causes the values of other channels to be relatively low, and finally a dark channel is generated.” – The sentence is not grammatically correct. Either remove “i.e”., or remove “and finally”. Possible solution:  Due to the lack of other colors, i.e., the color is concentrated in a certain channel, which causes the values of other channels to be relatively low, a dark channel is generated.
  • “the He’s algorithm” – without “the”
  • “as show in Algorithm …” – as SHOWN
  • “there are three region with the maximum difference in four regions” - three REGIONS
  • “The number of bright and dark pixels are denotes as Nb and Nd respectively” – are DENOTED as
  • “Finally, we find the maximum values Ab and Ad of dark channel in the light and dark areas respectively, and it is assumed that Ab and Ad are obtained at Y(n1) and Y(n2), the atmospheric value A is calculated as follows:” – this phrase should be split in at leas 2 separate sentences (for example, a second sentence could start with “The atmospheric value A is calculated as follows:”
  • “the value of atmospheric light A is estimate” – is ESTIMATED.
  • “It can be known from Eq(13), it is a linear transformation centered on k in the image I.” – sentence makes no sense. Possible solution: Eq(13) represents a linear transformation centered on k in the image I.
  • “First, use the rough transmittance map estimated by the dark channel prior algorithm as…” – First, WE use
  • “At the same time, there are also obvious color cast.” – “are” is in plural form and “cast” is in singular form
  • “As a result, various defogging algorithms based on sky region segmentation is emerging.” – ARE emerging (algorithms are in plural form)
  • Steps 2, 4, 5 from section 3.2.2.: the verbs should be in a different form: “Use” instead of “Using” (to be consistent with the other steps)
  • “Figure … is the defogging images achieved by two defogging algorithms” – Figure … ILLSUTRATES the DEFOGGED images obtained by two defogging algorithms
  • “At the same time, obvious edge effect appears” – At the same time, AN obvious edge effect appears.
  • “This will cause inaccurate the transmittance estimation and eventually cause edge effect.”. This will cause AN inaccurate transmittance estimation …
  • “Figure … are defogging images of four defogging algorithms.” – Figure …. ILLUSTRATES DEFOGGED images …
  • “by using defogging algorithm based the cost function [? ].” – based ON the cost function
  • “The reflective water surface and outside the lake of the second image show large distortion.” – it should be rephrased.
  • “This approach also make the transmittance estimation of the sky area” – also MAKES the transmittance …
  • “fogging image” – FOGGED image
  • “In the He’s defogging algorithm, he used the soft matting refine the transmittance.” – In He’s defogging algorithm, he used the soft matting TO refine the transmittance.
  • “and the reason is that our method has better defogging ability and more abundant detailed information is recovered than our methods.]” – I don’t understand what the authors meant by this phrase. It makes no sense.
